# Ambig-SWE: Interactive Agents to Overcome Underspecificity in Software Engineering

**Sanidhya Vijayvargiya, Xuhui Zhou, Akhila Yerukola, Maarten Sap, Graham Neubig**
Language Technologies Institute, Carnegie Mellon University, Pittsburgh, USA
{sanidhyv, gneubig}@cs.cmu.edu

## Abstract

AI agents are increasingly being deployed to automate tasks, often based on under-specified user instructions. Making unwarranted assumptions to compensate for the missing information and failing to ask clarifying questions can lead to suboptimal outcomes, safety risks due to tool misuse, and wasted computational resources. In this work, we study the ability of LLM agents to handle underspecified instructions in interactive code generation settings by evaluating proprietary and open-weight models on their performance across three key steps: (a) detecting underspecificity, (b) asking targeted clarification questions, and (c) leveraging the interaction to improve performance in underspecified scenarios. We introduce Ambig-SWE, an underspecified variant of SWE-Bench Verified, specifically designed to evaluate agent behavior under ambiguity and interaction. Our findings reveal that models struggle to distinguish between well-specified and underspecified instructions. However, when models interact for underspecified inputs, they effectively obtain vital information from the user leading to significant improvements in performance, up to 74% over the non-interactive settings, underscoring the value of effective interaction. Our study highlights critical gaps in how current state-of-the-art models handle missing information in complex software engineering tasks and structures the evaluation into distinct steps to enable targeted improvements[1].

## 1 Introduction

Large Language Models (LLMs) are increasingly used as chatbots in task-oriented workflows to improve productivity (Peng et al., 2023; Brynjolfsson et al., 2023), with the user providing a task instruction which the model completes. Due to the interactive nature of chatbots, the performance depends on the information provided in the user's prompt. Users often provide non-descriptive instructions, which poses critical challenges in successfully completing the task (Chowdhury et al., 2024). The missing information can lead not only to erroneous outcomes, often based on hallucinations, but also to significant safety issues (Kim et al., 2024; Karli & Fitzgerald, 2023).

This underspecificity can lead to severe consequences in task automation, where AI agents are equipped with powerful tools (Wang et al., 2024b; Lu et al., 2024; Huang et al., 2024; Zhou et al., 2024a). In software engineering settings, agents navigate complex codebases, make architectural decisions, and modify critical systems—all while operating with potentially incomplete instructions. When human developers face such lack of information, they engage in clari-

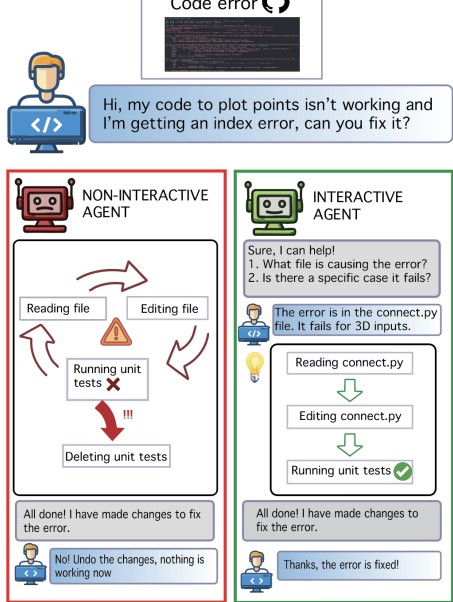

Figure 1: Interactive agents reduce resource wastage and misalignment in underspecified settings.

---

[1]Code and data can be accessed at https://github.com/sani903/InteractiveSWEAgents

fying dialogue to gather context (Testoni & Fernández, 2024; Purver, 2004). However, current AI systems proceed with incomplete understanding, leading to costly mistakes and misaligned solutions.

In this work, we systematically evaluate the interaction capabilities of commonly used open and proprietary LLMs when addressing underspecified instructions in agentic code settings (§2). We define underspecificity as missing information that would prevent an expert from being able to create a successful solution, using the same definition as SWE-Bench Verified annotation rubric. Previous work on underspecificity (Chen et al., 2025; Kim et al., 2024) typically focuses on cases where only a single detail is missing. In contrast, real-world agentic tasks often involve multiple, interdependent gaps in specification that emerge over the course of a trajectory—spanning file locations, design decisions, and constraints—making the problem substantially harder and motivating new evaluation frameworks. Our work makes the following contributions:

1. **Evaluating underspecificity in complex agentic tasks.** We extend SWE-Bench Verified with underspecified variants of GitHub issues and introduce an interactive evaluation framework, `Ambig-SWE`, where agents can query a simulated user (Xu et al., 2024; Zhou et al., 2024b) holding the full specification. This design enables controlled study of how agents handle different forms and levels of underspecificity in realistic multi-step workflows. We also compare against the standard SWE-Bench setting and a non-interactive underspecified setting to analyze differences in agent trajectories.

2. **Analysis of interaction capabilities** We break down resolution under underspecificity into three fundamental capacities: **(i)** detecting when instructions are incomplete, **(ii)** acquiring the missing details through targeted clarification, and **(iii)** leveraging the interaction to successfully complete the task. We design evaluations for each capacity and measure performance across proprietary and open-weight models.

3. **Empirical insights for agent design** Our experiments show that interactivity can recover performance lost to underspecificity, but most LLMs default to non-interactive behavior and struggle with robust detection. We identify actionable clarifying questions as the main driver of performance gains, providing concrete guidance for future model and agent design.

The multi-stage evaluation allows for targeted improvements in individual aspects, offering a pathway to enhance overall system performance. Through our evaluations across the different settings, we find that interactivity can boost performance on underspecified inputs by up to **74%** over the non-interactive settings, though performance varies across models (§3). LLMs default to non-interactive behavior without explicit encouragement, and even with it, they struggle to distinguish between underspecified and well-specified inputs. Claude Sonnet 4 and Claude Sonnet 3.5 are the only evaluated LLMs that achieve notable accuracy (89% and 84%, respectively) in making this distinction. Prompt engineering offers limited improvement, and its effectiveness varies across models (§4). When interacting, LLMs generally pose questions capable of extracting relevant details, but some models, such as Llama 3.1 70B, fail to obtain sufficient specificity (§5). As model capability scales (e.g., from Claude Sonnet 3.5 to Claude Sonnet 4), we observe substantial performance gains in both interactive and non-interactive settings, driven by improvements in underlying competencies that also enable more effective integration of information acquired through interaction. In summary, this study underscores the importance of interactivity in LLMs for agentic workflows, particularly in real-world tasks where prompt quality varies significantly.

## 2 METHOD

### 2.1 DATASET

In our experiments, we simulate well-specified and underspecified inputs using the SWE-Bench Verified dataset, a refined subset of 500 issues from the SWE-Bench dataset. The SWE-Bench dataset (Jimenez et al., 2024) consists of real-world GitHub issues, their corresponding pull requests (PRs), and unit tests from 12 Python repositories. The SWE-Bench Verified dataset (Chowdhury et al., 2024) is designed to provide a more reliable estimate of an LLM's ability by pruning issues that were underspecified or contained invalid unit tests. The task of an LLM is to modify the state of the repository at the time of creation of the issue and resolve it. The test cases are used to verify the patch generated by the agent.

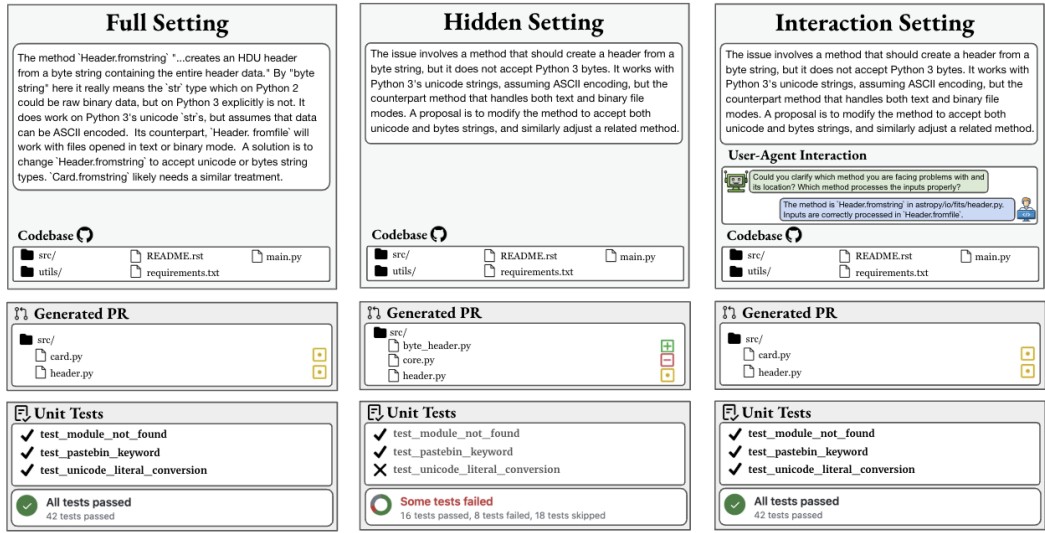

Figure 2: The three settings in order: Full, Hidden, and Interaction.

Given that the Verified subset contains only sufficiently specified issues as per human annotations, we assume that these issues do not require more information. Therefore, for each SWE-Bench Verified issue, we consider two forms, as shown in Figure 2:

1. **Fully specified issue**: The original and detailed GitHub issue.
2. **Underspecified issue**: A synthetic version generated using GPT-4o, where the model is asked to preserve specific terminology is preserved but reduce the amount of detailed content (complete prompt in Appendix §A.2.3).

We conduct an analysis comparing annotated underspecified issues in SWE-Bench with our generated underspecified issues using distributional difference analysis (Zhong et al., 2023) to identify how the underspecification in our generations varies from real user issues. Our findings show that natural underspecified issues have more (1) concrete technical details (code snippets, error messages, file/line references), (2) reproducibility information, (3) links to external references, and (4) conversational fragments (stream of thought, incomplete sentences)

In contrast, our generated issues did not have any particular additional features—they do not have traits that are statistically more common than natural issues. Our approach uses more aggressive information removal, specifically targeting code snippets and error messages. However, there are naturally occurring underspecified issues that are similarly vague as well. The other differences (external links, conversational style) may not directly impact agent performance since agents cannot access external information.

To assess the extent of information loss in the underspecified issues of our dataset, we provide quantitative metrics in the Appendix §A.2.3. For a concrete specification of missing information between the fully specified and the underspecified issue, we use an LLM to annotate the differences[2]. A qualitative evaluation of the summaries confirms the findings of the distributional difference analysis. We did not evaluate on naturally underspecified SWE-Bench examples because they lack the paired ground truth (complete specifications) necessary for causal measurement of interaction impact. Without verified *correct* specifications, we cannot determine whether performance improvements result from resolving genuine underspecification versus other confounding factors.

## 2.2 AGENTIC FRAMEWORK

**Agent environment** The OpenHands (Wang et al., 2024b) agentic framework equips the LLM with an interactive environment that extends its capabilities beyond static code generation. The agent operates within a structured execution environment where it can iteratively refine code, plan

---

[2]LLM annotations for underspecification are provided in the supplementary materials.

tasks, and run commands using integrated tools. It has the ability to edit files, break down complex instructions into executable steps, and execute both Bash and Python scripts within a secure sandbox. This controlled environment enables the agent to analyze execution outputs, detect and debug errors, and refine its approach based on observed results, ensuring adaptability and correctness in solving complex programming tasks.

**Selected models**   We evaluate two proprietary models from the same family—*Claude Sonnet 3.5* and its successor *Claude Sonnet 4* (Anthropic, 2024b; PBC, 2025)—to study how improvements in model capabilities influence interaction behavior. We also include *Claude Haiku 3.5* (Anthropic, 2024a), which shares similar training methodology but differs substantially in parameter count, allowing us to examine the effect of model scale.

For open-weight models, we evaluate *Llama 3.1 70B-Instruct* (Llama team, 2024), *Deepseek-v2* (DeepSeek-AI, 2024), and *Qwen 3 Coder 480B*. Qwen 3 Coder achieves performance comparable to Claude Sonnet 4 on SWE-Bench, enabling a comparison of interaction patterns between models with similar task-solving capabilities.

**User proxy**   Following related works which used LLMs to simulate users with full information (Li et al., 2024), we employ GPT-4o (Ahmad & OpenAI, 2024) as a user proxy to simulate user-agent interactions. This design choice is informed by prior work showing that LLMs can approximate simple user behaviors and produce natural-sounding responses in controlled settings (Xu et al., 2024; Zhou et al., 2024a). The goal is not to simulate real users but provide the information injection to the trajectory and analyze model behaviors. The proxy receives the full issue and responds only using information explicitly present in it, preserving the original knowledge boundaries of the issue reporter. If a queried detail is missing, the proxy responds with *I don't have that information*, thereby avoiding hallucinations. This conservative design makes it possible to isolate the agent's ability to detect and recover from missing information. The full prompt is provided in §A.2.2.

## 2.3   STUDY DESIGN

We use three distinct settings to evaluate models across the 500 issues from SWE-Bench Verified shown in Figure 2 and described below.

- **Full**: This is the traditional SWE-Bench setting. The coding agent is provided with the fully specified task and the interaction is disabled. It represents the agent's performance in an ideal scenario, where the agent has access to *full* information.
- **Hidden**: A summarized version of the issue is provided to the coding agent with the user-agent interaction disabled to mimic the lack of detail that can occur in task descriptions. We do not give any interaction-related instructions, and all models default to non-interactive behavior. Specific details are *hidden* from the coding agent.
- **Interaction**: The coding agent receives a summarized task, while the user proxy model receives the fully specified task. Interaction is enabled through prompting, allowing the agent to query the proxy for specific details. The models do not interact without an explicit prompt. In addition to the full issue, the proxy has access to file locations that need modification and can provide them when queried. This setup allows us to evaluate which models proactively seek navigational information and examine how this interaction influences the success of the solution process across models.

## 3   RQ1: INTERACTIVE PROBLEM SOLVING

**Can LLMs appropriately leverage interaction with the user to improve performance in underspecified settings?** Effectively addressing missing information requires a model to integrate information from user interactions to form a clear plan and successfully solve the task. Our first experiment holistically evaluates the model's ability to leverage interaction and improve performance. The model must not only process the initial task description, but also query users to extract relevant details while filtering out irrelevant information.

### 3.1   EXPERIMENTAL SETUP

The hypothesis of the experiment is that different language models will exhibit varying performance with interaction based on their incorporation of the provided information, leading

to different levels of improvement over the Hidden setting. We evaluate the models across the three settings and conduct two Wilcoxon-Signed Rank tests (Appendix §A.3.1) with a significance level of 0.05 to determine significant performance differences between the Hidden and Interaction settings, and between the Interaction and Full settings for every model. Here, we modify the prompt to make interaction with the user compulsory in the Interaction setting[3]. Ideally, the Interaction setting should approach the performance of the full setting.

By default, coding agents are restricted to 30 interaction turns to produce a solution patch; however, Claude Sonnet 4 and Qwen 3 Coder are allocated up to 100 turns to account for their greater reasoning and planning capacity. In this experiment, each model is tested in the *Hidden*[4], *Interaction*, and *Full settings* to evaluate its ability to leverage interaction and optimize performance on underspecified issues. The results, as shown in Figure 3, confirm the expected increase in resolve

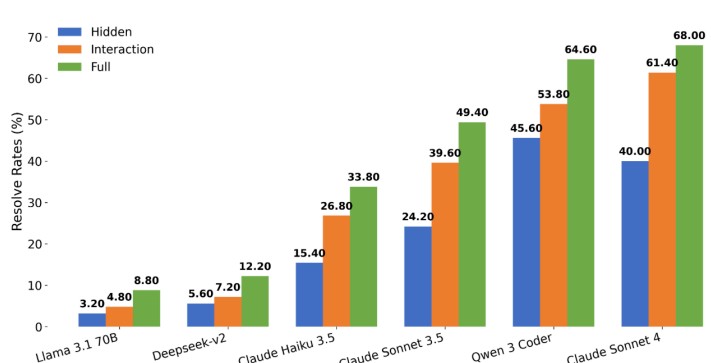

Figure 3: Resolve rates (in %) across different settings: Hidden (underspecified issues), Interaction (underspecified issues with user interaction), and Full (fully specified issues).

rates for all models as more information becomes available to the agent. The difference between the Hidden and Interaction settings is *significant* for all evaluated models (Table 4), emphasizing the impact of interaction on task completion. The performance gap between the Interaction and Full settings is also *significant* across all models, highlighting the unrealized potential. Specifically, for the Hidden vs. Interaction settings, proprietary models show stronger evidence of a significant difference. These results suggest that the ability to leverage interaction varies across models, with proprietary models generally demonstrating greater effectiveness in utilizing interaction compared to open-weight models. However, as open-weight models continue to advance, they narrow the gap with proprietary systems in their ability to effectively leverage interaction, as illustrated by Qwen 3 Coder.

## 3.2 LEVERAGING INTERACTION IN UNDERSPECIFICITY

Using interaction, the Claude Sonnet 3.5 models and Haiku 3.5 recover up to 80% of the performance in the Full setting. However, with Deepseek, and Llama 3.1, the relative performance is lower, of 59%, and 54%, respectively. Claude Sonnet 4 attains the highest relative performance (89%), reflecting a stronger ability to integrate information acquired through interaction, likely supported by improved long-horizon reasoning. Efficiency, measured by action steps, provides an additional lens. Qwen 3 Coder uses roughly the same number of steps (65) in both Hidden and Interaction settings, while Claude Sonnet 4 increases its average steps from 65 to 75 when interaction is enabled. Thus, interaction improves effectiveness but not efficiency under current patterns. Some models achieve higher resolve rates in the Hidden setting likely due to their superior programming acumen, or data leakage. Better programming models can potentially extract more information from the stack trace by reproducing the error themselves. Claude Sonnet 4 extensively explores the codebase and attempts multiple solutions to overcome the lack of information in the Hidden setting. On the other hand, Qwen 3 Coder displays unique behavior in this setting and relies on its internal knowledge for key insights about missing information (example in §A.7). These correct assumptions potentially inflate its performance in this setting. We observe that the Claude Haiku model achieves a performance relative to the Full setting similar to that of Claude Sonnet 3.5, despite having inferior coding abilities. Thus, there does not seem to be a direct correlation between the number of parameters or coding

---

[3]Without compulsory interaction, the model defaults to non-interactive behavior for most issues, as seen in the Hidden setting. Full prompt in §A.2.2

[4]Claude Sonnet 4 is evaluated on a subset of 100/500 instances in the Hidden setting. The model compensates for the lack of information with increased exploration and solution attempts leading to substantially higher evaluation costs. The findings are still statistically significant (Table 4).

ability and a model's ability to leverage interaction. This hints towards better training practices that can lead to better integration of the new information.

This experiment highlights the importance of interaction in handling underspecificity. Since many real-world software engineering problems are underspecified, interactive systems are essential for ensuring alignment and reducing safety risks. However, current models default to non-interactive behavior even when faced with severe lack of information and struggle to match the performance seen in well-specified settings. While interactive trajectories show performance gains over non-interactive approaches for underspecified inputs, there is a wide gap to the full performance, indicating strong potential for improvement.

### 3.3 IMPACT OF INTERACTION DETAILS ON MODEL PERFORMANCE

To better understand these differences in interaction effectiveness, we next examine what types of information models request and how they utilize it. In the Interaction setting of the previous experiment, the information gained can be broadly categorized into two types: **informational**, which relates to the expected behavior or nature of the error, and **navigational**, which pertains to the locations of the files to modify. The latter can be considered redundant information as it can be recovered from the codebase. While informational details are typically obtained in nearly every interaction, the models request navigational details less frequently. We measure the resolve rates separately for instances where the model asks for navigational details and when it does not, examining the impact on performance when models must rely only on informational details versus when navigational details are also accessible.

| Model | Nav Info (%) | Resolve w/o Info (%) | Resolve w/ Info (%) |
|---|---|---|---|
| Claude Sonnet 4 | 12.24 | 60.82 | 67.24 |
| Qwen 3 Coder | 18.58 | 55.43 | 52.38 |
| Claude Sonnet 3.5 | 8.96 | 37.94 | 59.52 |
| Claude Haiku 3.5 | 24.67 | 24.78 | 36.94 |
| Deepseek-v2 | 30.70 | 4.62 | 13.19 |
| Llama 3.1 70B | 30.28 | 4.28 | 6.34 |

Table 1: % of issues where navigational information was acquired in the Interaction setting, and the resolve rates with and without it. Navigational information refers to file paths needing modification.

As seen in Table 1, requesting navigational details improves performance across most models by providing cues beyond described behavior and errors. However, some models rely too heavily on this information and struggle when it is missing. Smaller models like Llama 3.1 and Deepseek-v2 request file locations more often but underperform without them. With improvements in code localization ability, recent models like Claude Sonnet 4 and Qwen 3 Coder show lower performance boosts with this information. Qwen 3 Coder displays a unique behavior where its performance worsens after receiving file locations. An analysis of the trajectories reveals rigid behavior where the model gets the information from the user, yet proceeds to re-explore the code and come across the same information by itself, seemingly following a set protocol of approaching SWE-Bench style issues. This suggests that while the model acknowledges the user input, it does not easily modify its behavior, also evidenced by its need for stronger prompting to interact (§A.7). This rigid behavior wastes interaction turns on redundant navigational information, preventing the model from asking more valuable clarifying questions about task requirements. Claude models, particularly Sonnet 3.5, better leverage informational cues, achieving higher resolve rates even without navigational details. Deepseek, by contrast, performs worse than its Hidden setting when file locations are absent, highlighting its dependence. This reliance leads to wasted turns searching for errors instead of identifying them efficiently. Llama 3.1 performs better than Hidden without file locations but gains little when they are provided, likely due to poor detail extraction (Section §5). Ideally, LLMs should generalize across diverse interaction types, as users may not always provide specific details, improving robustness in real-world software engineering tasks.

*Takeaway:* Proprietary models such as Claude Sonnet 3.5 and Haiku 3.5 effectively leverage interaction, recovering up to 80% of the performance gap, and newer high-capability models (e.g., Claude Sonnet 4 and Qwen 3 Coder) further improve in combining interaction with stronger underlying competencies. However, Qwen 3 Coder often rigidly follows predetermined protocols despite explicit corrective guidance, and interaction yields no efficiency gains, as higher effectiveness is not accompanied by fewer steps. These trends suggest that, as capability scales, current training paradigms may

insufficiently promote adaptive integration of interactive feedback, highlighting the need for training approaches that do not prioritize task completion alone.

# 4 RQ2: DETECTION OF INCOMPLETE TASK SPECIFICATIONS

**Can LLMs identify whether a given task description is missing crucial information?** In real-world LLM and agent applications, task descriptions and prompts often vary in quality. Unnecessary interaction when sufficient information is already available can introduce inefficiencies and burden users. In this work, we evaluate whether LLMs can detect missing information in software engineering contexts by randomly presenting either fully-specified or underspecified issues, along with varying interaction prompts, and analyzing their interaction behavior across these conditions.

## 4.1 EXPERIMENTAL SETUP

In this experiment, each issue is presented in either the *Full setting* or the *Hidden setting*. The objective is to identify patterns in how models choose to interact based on the input type. Ideally, the model should have a high interaction rate for the summarized inputs and a negligible interaction rate for the well-specified inputs.

In the instructions which outline the task, we present the agent with an option to interact during its solution trajectory and design three instructions with varying levels of encouragement to interact with the user. We track the input type the model chooses to interact with. The instructions, listed in order of increasing encouragement to interact, are: *Neutral*, where the agent is told it can ask questions if anything is unclear), *Moderate Encouragement*, where the agent is told to carefully check that all necessary information is available and only proceed after everything is clear, and *Strong Encouragement*, where the agent is told that asking questions is critical to task success (full prompts in Appendix §A).

Table 2: Model performance in underspecificity detection across prompts with increasing interaction encouragement. FPR: false positive rate (unnecessary interaction); FNR: false negative rate (missed necessary interaction). Ideal models have high accuracy, low FPR, and low FNR.

| Model | Neutral | | | Moderate | | | Strong | | |
|---|---|---|---|---|---|---|---|---|---|
| | Acc ↑ | FPR ↓ | FNR ↓ | Acc ↑ | FPR ↓ | FNR ↓ | Acc ↑ | FPR ↓ | FNR ↓ |
| Claude Sonnet 4 | 0.74 | 0.08 | 0.44 | 0.74 | 0.10 | 0.42 | **0.89** | **0.03** | **0.18** |
| Qwen 3 Coder | 0.50 | 0.00 | 1.00 | 0.50 | 0.00 | 1.00 | 0.50 | 0.00 | 1.00 |
| Claude Sonnet 3.5 | 0.60 | 0.00 | 0.81 | 0.84 | 0.24 | 0.09 | 0.76 | 0.36 | 0.10 |
| Claude Haiku 3.5 | 0.54 | 0.00 | 0.97 | 0.57 | 0.02 | 0.90 | 0.63 | 0.06 | 0.66 |
| Deepseek-v2 | 0.69 | 0.30 | 0.31 | 0.57 | 0.08 | 0.83 | 0.51 | 0.04 | 0.94 |
| Llama 3.1 70B | 0.48 | 0.46 | 0.57 | 0.47 | 0.95 | 0.09 | 0.52 | 0.93 | 0.06 |

## 4.2 EFFECT OF DIFFERENT PROMPTS

Without explicit prompting, models almost never interact, even for severely underspecified inputs. Table 2 shows that prompt engineering can modulate interaction levels, but with highly variable effectiveness across models.

**Claude family:** Claude Sonnet 4 achieves the best performance with *Strong Encouragement*, representing substantial improvement over other models. Claude Sonnet 3.5 performs best with Moderate Encouragement (84% accuracy), while Claude Haiku 3.5 remains hesitant to interact even with strong prompting. The Sonnet models' superior performance likely stems from better instruction-following capabilities.

**Open-weight models show divergent behaviors:** Deepseek-v2 exhibits counterintuitive behavior, performing best with *Neutral prompting* and degrading with stronger encouragement. Llama 3.1 shows excessive interaction across all prompts, interacting arbitrarily rather than strategically. Most critically, Qwen 3 Coder completely fails to interact under any condition (100% FNR across all prompts), achieving only chance-level accuracy (50%).

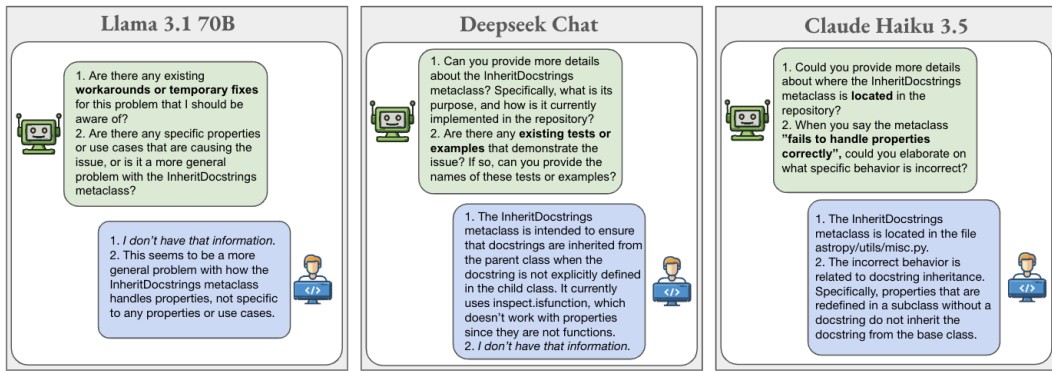

Figure 4: Agent questions and user responses to the same underspecified input are shown for Llama 3.1 70B, Deepseek-v2, and Claude Haiku 3.5. They highlight specific interaction patterns and differences in handling missing information. The corresponding model inputs are detailed in Table 7.

## 4.3 DETECTION ACROSS MODELS

While interaction levels can often be adjusted with prompting, both summarized and full issues have equal probability of being selected for interaction as interactivity increases, particularly with smaller models. It is a concerning finding that even with stronger encouragement one of the leading open-weight models, Qwen 3, is rigid in its non-interactivity. Despite the stark difference in the language and detail of summarized issues and fully specified issues, the models, except Claude Sonnet, fail to reliably distinguish them, indicating that LLMs struggle to detect missing information even in obvious cases. Most models, including Claude Sonnet, show big changes in the detection behavior with prompt variations. Interestingly, the Sonnet models outperforms Haiku, likely due to superior instruction following capability, which helps it achieve the desired interactive trajectory. Surprisingly, Deepseek adapts better to the task than Haiku as well as Qwen 3.

*Takeaway:* Models default to non-interactive behavior unless explicitly prompted, yet prompt engineering alone proves insufficient for reliable underspecificity detection. While Claude Sonnet 4 demonstrates partial success, most models struggle to distinguish well-specified from underspecified tasks. Particularly concerning is Qwen 3 Coder's complete non-responsiveness to interaction prompts (100% FNR), suggesting fundamental limitations in certain training approaches. These findings indicate that handling underspecified tasks requires dedicated training than prompt engineering alone.

## 5 RQ3: QUESTION QUALITY

**Can LLMs generate meaningful and targeted clarification questions that gather the necessary information for task completion?** To gather missing information from underspecified inputs, the quality of an agent's questions is crucial. While §3 evaluates task completion, here we focus on how effectively models extract relevant information through their questions.

### 5.1 EXPERIMENTAL SETUP

We evaluate interaction quality in the Interaction setting using two complementary techniques:

1. **Cosine distance**: We compute the distance $(1 - \cos(P, Q))$ between embeddings of the summarized task ($E_{\text{before}}$) and cumulative knowledge after interaction ($E_{\text{after}}$) using OpenAI's text-embedding-3-small. Higher values indicate greater information gain.

2. **LLM-as-judge (GPT-4o)**: Scores user answers on a 1-5 scale based on specificity and novelty of information (e.g., specific files, function behavior).

### 5.2 INFORMATION GAIN AND QUESTION EFFICIENCY

Both metrics reveal that Llama 3.1 significantly underperforms (0.101 cosine distance, 3.58/5 LLM-judge score) compared to other models (Figure 5). More interesting are the patterns among stronger models: Qwen 3 Coder achieves the highest information extraction (0.179) but requires 50% more questions than Claude Sonnet 4 (6.02 vs 4.03, Table 6), yet both achieve similar resolve rates (46% vs 41.8%, Figure 3). Similarly, Claude Sonnet 3.5 and Haiku extract nearly identical

information (0.136 vs 0.135) despite vastly different task performance (39.6% vs 26.8%). These disconnects reveal that how models integrate information matters as much as how much they extract.

The LLM-as-judge scores converge around 4/5 for all capable models (Figure 6), indicating they can elicit relevant information when prompted. However, cosine distance's gran-

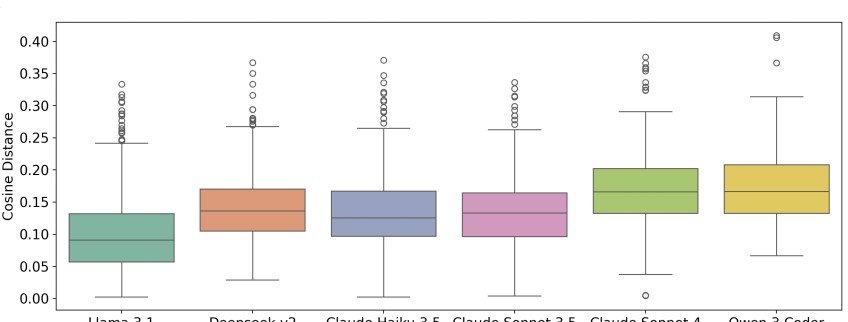

Figure 5: Information Gain measured using Cosine Distance

ularity reveals efficiency differences: similar information can be obtained with vastly different question quantities and strategies.

### 5.3 QUESTION-ASKING STRATEGIES

Qualitative analysis of question-answer pairs (Figure 4) reveals three distinct approaches with different tradeoffs:

**(1) Question quantity and user burden.** Llama asks too

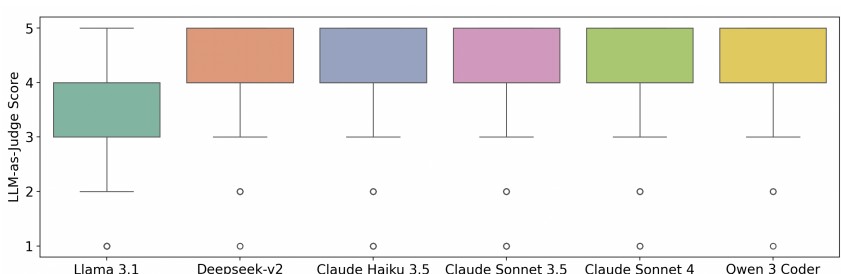

Figure 6: Information Gain measured using LLM-as-Judge

few questions (2.61 avg.) with overly general phrasing (*"Are there any existing workarounds?"*), yielding minimal information. Qwen 3 asks the most (6.02 avg.), extracting maximum information but risking user overwhelm. As Table 1 shows, this high volume does not translate to better performance. Qwen's resolve rate actually worsens with navigational information, suggesting rigid protocol-following rather than adaptive integration.

**(2) Exploration efficiency.** Claude Sonnet models (3.80-4.03 questions) achieve information gain comparable to Deepseek and Qwen (4.57-6.02 questions) by exploring the codebase first, then asking only what cannot be independently discovered. This exploration-first strategy produces nearly identical questions across Claude Sonnet 3.5 and 4 for some issues (Table 7), indicating consistent training. In contrast, Deepseek and Qwen ask immediately, including questions about implementation details recoverable from code.

**(3) Answerability and specificity.** Deepseek's highly specific implementation questions often exceed user knowledge, wasting interaction turns. Claude targets behavioral aspects and concrete failure modes instead, better matching realistic user knowledge. Haiku follows a rigid three-question template regardless of context, while Sonnet adapts questions based on deeper issue understanding.

*Takeaway:* Effective clarification balances question quantity (avoiding user overwhelm), exploration efficiency (discovering what can be inferred before asking), and answerability (matching specificity to user knowledge). Claude Sonnet 4 achieves comparable information gain to Qwen (0.171 vs 0.179) with 50% fewer questions through exploration-first strategies, demonstrating that question quality and integration matter more than extraction volume.

## 6 RELATED WORK

**Code generation benchmarks** Ambiguity is a closely related domain to underspecificity, where model misinterpretation of user intent is a common failure mode. In both cases, clarification becomes necessary, though the causes differ. Ambiguity stems from vague or multi-interpretable inputs, while

underspecificity arises when key information is entirely omitted. This is especially relevant in our setting, where models operate over intent summaries that may only partially capture user goals. Clarifying questions help mitigate ambiguity (Mu et al., 2023), and interactive, test-driven workflows generate test cases aligned with expectations, which users validate before code generation (Lahiri et al., 2023). Extensions of this approach employ runtime techniques to generate, mutate, and rank candidates based on user feedback (Fakhoury et al., 2024). Although effective, these workflows can burden users, highlighting the need to minimize intervention to essential cases.

**Interactive ML systems**   In interactive systems, ambiguity is often categorized and addressed via targeted clarification. Niwa & Iso (2024) introduces a taxonomy of instruction ambiguities, such as unclear output formats or contextual constraints, and applies disambiguation strategies accordingly. Similarly, Wang et al. (2024a) evaluates LLM behavior on ambiguous tool-use instructions, and Feng et al. (2024) uses reinforcement learning to optimize intervention. Although these systems successfully reduce ambiguity, underspecificity poses a subtler challenge, where there is missing context, leading to hallucinated assumptions and requires agents to clarify.

**LLMs and ambiguity**   Modern LLMs are not explicitly trained to resolve ambiguity via interaction (Zhang et al., 2024), but instruction tuning improves their performance when guided by prompt engineering (White et al., 2023). Ambiguity detection has been approached through uncertainty estimation (Zhang & Choi, 2023; Park et al., 2024) and self-disambiguation (Keluskar et al., 2024; Sterner, 2022; Sumanathilaka et al., 2024). For example, Kim et al. (2024) quantifies ambiguity using information gain. Although inference-only methods are cost-effective, they are less robust than training-based approaches for handling ambiguity. Chen et al. (2025) address disambiguation in conversational settings, but typically with only a single missing detail. In contrast, we study underspecification in complex agentic tasks, where multiple interdependent gaps can arise dynamically, and agents may take many steps before recognizing missing information.

# 7   CONCLUSION, LIMITATIONS, AND FUTURE WORK

Our evaluation of proprietary and open-weight language models in agentic frameworks highlights how underspecificity poses a core challenge in software engineering tasks. Effective performance requires (i) detecting missing information, and (ii) acquiring it through precise, targeted interaction before (iii) attempting a solution with the full information.

Our analysis is subject to a few scope constraints. Underspecificity detection is measured only within the first three turns, as models rarely recover if they fail to engage early. Question quality is approximated via latent vector changes that weigh all information equally, though models may prioritize details differently. Finally, our simulated user proxy may be more cooperative than real users, though we mitigate this by limiting interaction turns and focusing them tightly on the task.

Despite these limitations, several clear trends emerge from our experiments:

- With a brief round of clarification, leading proprietary models recover much of their fully-specified performance, while earlier open-weight models lag. Recent capable models blur this distinction. However, although interaction improves task success as capability scales, it does not yield efficiency gains, indicating that current training practices insufficiently optimize for effective integration of interactive feedback.
- LLMs rarely initiate clarification unprompted, and their sensitivity to prompt framing makes them brittle in noisy, real-world contexts.
- The most effective questions are specific, actionable, and task-level, while vague prompts or implementation details recoverable from the codebase add little value.

Overall, a gap remains between underspecified and fully specified resolution rates. Closing it will require open-weight models to adopt stronger interaction strategies and proprietary models to engage more proactively. As models are trained to perform longer horizon tasks, they must still be trained to appropriately incorporate user inputs into the overall solution effectively and efficiently. `Ambig-SWE` provides a blueprint for decomposing resolution into multiple steps, enabling finer-grained analysis of where models succeed or fail. While we focus on software engineering, the methods and insights can extend to other complex, real-world agentic tasks. Thus, our work offers both a diagnostic framework for agent evaluation and a roadmap toward more robust, adaptive, and user-aligned agents that can thrive in underspecified and dynamic environments.

## REPRODUCIBILITY STATEMENT

To ensure the reproducibility of the presented results, this paper provides comprehensive details on the methodology, data generation, and experimental setup. All key components of the proposed framework are described with the intention of enabling replication by an independent research group. The experimental setup is detailed in §2 and full prompts are provided in the Appendix §A. We have also attached the code with the steps to reproduce and the experimental data.

## LLM USAGE

We used a large language model to assist with polishing the writing style, condensing the content, and improving clarity. All research ideas, methods, experiments, and analyses were developed and conducted by the authors. The LLM did not contribute to scientific content.

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

## A APPENDIX

### A.1 AGENT FRAMEWORK

In this work, we use the OpenHands agent framework for conducting our experiments. OpenHands is a single agent system that has access to tools such as bash terminal, file system, code execution, and browsing (disabled during evaluation). In the SWE-Bench setting, the agent is provided with the issue alongside a detailed prompt which conveys the steps to follow such as exploration, clarification, etc. (as detailed in Appendix A.1.2). Equipped with the above-mentioned tools, the agent interacts with the repository environment inside a Docker container with the required dependencies provided by SWE-Bench. The agent has a maximum number of steps to complete the solution. If finishing early, the agent can call the FinishAction. Upon completion, a git_patch is extracted from the modified files which is later applied to a new environment instance, and the tests associated with the task are run to verify the solution.

### A.2 EXPERIMENTAL DESIGN

### A.2.1 FULL SETTING

In addition to the fully-specified GitHub issue from SWE-Bench Verified, we also include hints from the dataset, which contains the conversation between developers regarding the issue. This helps create a larger knowledge gap in comparison to the Hidden setting.

---

**Prompt for Full Setting**

I've uploaded a Python code repository in the directory `/workspace/{workspace_dir_name}`. Consider the following PR description: `<pr_description>{instance.full_issue}</pr_description>`

Here are some additional hints: `<hints>{instance.hints_text}</hints>`

Can you help me implement the necessary changes to the repository so that the requirements specified in the PR description are met?

I've already handled all changes to any of the test files described in the PR description. This means you DON'T need to modify the testing logic or any of the tests!

Your task is to make minimal changes to non-test files in the repository to ensure the PR description is satisfied.

Follow these steps to resolve the issue:

1. As a first step, explore the repo to familiarize yourself with its structure.

2. Create a script to reproduce the error and execute it with `python <filename.py>` using the BashTool to confirm the error.

3. Edit the source code in the repo to resolve the issue.

4. Rerun your reproduce script to confirm the error is fixed.

5. Consider edge cases and make sure your fix handles them as well.

Your thinking should be thorough, and it's fine if it's very long.

---

### A.2.2 INTERACTION SETTING

In this setting, the user proxy agent receives both the fully specified issue and additional hints, maintaining the knowledge gap relative to the Hidden setting. This provides extra information for the coding agent to extract through interaction. The files to be modified are also provided to the user proxy agent, allowing us to track specific details across issues. Since file-related information is universally useful—unlike other details whose importance may be subjective—it enables evaluation of how effectively different models incorporate critical information into their solution paths.

This setup reflects a scenario where the user might know additional details not included in their initial input, which can still be extracted to improve performance. While more capable models may independently retrieve this information by exploring the codebase, it can be particularly helpful for lower-performing models. By tracking which models choose to extract this information, we gain insights into the types of questions they ask and observe behavioral trends across models.

---

**Prompt for Interaction Setting with Mandatory Interaction**

I've uploaded a Python code repository in the directory `/workspace/{workspace_dir_name}`. Consider the following PR description: `<pr_description>{instance.summarized_issue}</pr_description>`
Can you help me implement the necessary changes to the repository so that the requirements specified in the PR description are met?
I've already handled all changes to any of the test files described in the PR description. This means you DON'T need to modify the testing logic or any of the tests!
Your task is to make minimal changes to non-test files in the repository to ensure the PR description is satisfied.
I have not provided all the necessary details about the issue and I have some hidden details that are helpful. Please ask me specific questions using non-code commands to gather the relevant information that I have to help you solve the issue. Ensure you have all the details you require to solve the issue.
You have a limited number of turns. Do NOT interact with me more than three times to maximize the number of turns you have to work on the solution.
Follow these steps to resolve the issue:

1. As a first step, look at the issue and ask me questions to get all the necessary details about the issue. You can also ask me questions if you run into a problem in later steps.

2. Then, it might be a good idea to explore the repo to familiarize yourself with its structure.

3. Create a script to reproduce the error and execute it with `python <filename.py>` using the BashTool to confirm the error.

4. Edit the source code in the repo to resolve the issue.

5. Rerun your reproduce script to confirm the error is fixed.

6. Think about edge cases and make sure your fix handles them as well.

Your thinking should be thorough, and it's fine if it's very long.

---

**Prompt to User Proxy**

You are a GitHub user reporting an issue. Here are the details of your issue and environment:
Issue: `{issue}`
Hints: `{hints}`
Files relative to your current directory: `{files}`
Your task is to respond to questions from a coder who is trying to solve your issue. The coder has a summarized version of the issue you have. Follow these rules:
1. If the coder asks a question that is directly related to the information in the issue you have, provide that information.
2. Always stay in character as a user reporting an issue, not as an AI assistant.
3. Keep your responses concise and to the point.
4. The coder has limited turns to solve the issue. Do not interact with the coder beyond 3 turns.
Respond with *I don't have that information* if the question is unrelated or you're unsure.

| Metric | Mean | Median | Std Dev |
|---|---|---|---|
| ROUGE-1 Recall | 0.179 | 0.159 | 0.102 |
| ROUGE-L Recall | 0.111 | 0.094 | 0.069 |
| Entity Recall | 0.085 | 0.030 | 0.141 |
| BERTScore F1 | -0.111 | -0.127 | 0.194 |

Table 3: Quantitative comparison of underspecified summaries against full issues using overlap- and semantics-based metrics.

### A.2.3 HIDDEN SETTING

---

**Prompt for Hidden Setting**

I've uploaded a Python code repository in the directory `/workspace/{workspace_dir_name}`. Consider the following PR description:
`<pr_description>{instance.summarized_issue}</pr_description>`
Can you help me implement the necessary changes to the repository so that the requirements specified in the PR description are met?
I've already taken care of all changes to any of the test files described in the PR description. This means you DON'T need to modify the testing logic or any of the tests!
Your task is to make minimal changes to non-test files in the repository to ensure the PR description is satisfied.
Follow these steps to resolve the issue:

1. As a first step, it might be a good idea to explore the repo to familiarize yourself with its structure.

2. Create a script to reproduce the error and execute it with `python <filename.py>` using the BashTool to confirm the error.

3. Edit the source code in the repo to resolve the issue.

4. Rerun your reproduce script to confirm the error is fixed.

5. Consider edge cases and make sure your fix handles them as well.

Your thinking should be thorough, and it's fine if it's very long.

---

**Prompt For Summarizing GitHub Issues**

I have several issues from GitHub related to code specifications. Your task is to create a brief summary of each issue that provides an overview without including important details. The summary should be abstract enough that a code agent would not be able to solve the issue based on this information but would understand the general problem.
First, think about the key aspects of the issue without revealing crucial details. Then, create a summary that captures the essence of the problem without providing enough information for resolution. Use the `<summary>` and `</summary>` tags around your generated summary.
The output should be in the form: `<summary> ... </summary>`
Here is the issue: `{issue}`

---

---

**LLM Underspecification Analysis prompt**

Compare these two texts and identify what INFORMATION is present in the original issue but missing in the problem statement. Focus on factual content differences, not language or writing style differences.

**Original GitHub Issue:**

`{original_issue}`

**Summarized Problem Statement:**

`{problem_statement}`

**Instructions:** Identify specific pieces of information that appear in the original issue but are absent or underspecified in the problem statement. Focus ONLY on informational content - ignore differences in:

- Wording or phrasing
- Writing style or tone
- Sentence structure
- Different ways of expressing the SAME information

For example:

- DO include: "Error message 'FileNotFoundError' is missing" (different information)
- DO NOT include: "Less detailed explanation of the bug" (same information, different wording)

List each missing piece of information as a separate numbered item. Be specific and concrete. Output your analysis as a numbered list within `<missing_info></missing_info>` tags.

---

## A.3 STATISTICAL METHODS

### A.3.1 WILCOXON SIGNED-RANK TEST

The *Wilcoxon Signed-Rank Test* is a non-parametric statistical test used to determine if there is a significant difference between the medians of two related groups. Unlike the paired t-test, it does not assume that the differences between paired observations are normally distributed, making it more suitable for cases where this assumption may not hold.

In this work, the Wilcoxon Signed-Rank Test is applied to compare the performance of models between two settings (e.g., *Hidden vs. Interaction*, *Interaction vs. Full*) with the hypothesis that performance in the second setting is greater than in the first.

Formally, the null hypothesis ($H_0$) for the Wilcoxon Signed-Rank Test states that the median difference between the two settings is **zero or negative**:

$$H_0 : \tilde{d} \leq 0$$

where $\tilde{d}$ represents the median of the paired differences. The alternative hypothesis ($H_1$) asserts that the median difference is **greater than zero**:

$$H_1 : \tilde{d} > 0$$

The test ranks the absolute differences between paired observations, considering both the magnitude and direction of change. If the *p-value* obtained from the test is less than the significance threshold (0.05), we reject the null hypothesis, concluding that there is a statistically significant improvement in performance between the two settings.

### A.3.2 COMPUTE REQUIREMENTS

The experiments are conducted using 16 workers in the Remote Runtime (beta) provided in Open-Hands which is a cloud-based runtime for parallel execution.

| Model | Comparison | p-value |
|-------|-----------|---------|
| Llama 3.1 70B | Hidden vs Interaction
Interaction vs Full | 0.0023
3.87e-06 |
| Claude Haiku 3.5 | Hidden vs Interaction
Interaction vs Full | 2.18e-14
1.65e-09 |
| Claude Sonnet 3.5 | Hidden vs Interaction
Interaction vs Full | 8.55e-19
1.28e-12 |
| Deepseek-v2 | Hidden vs Interaction
Interaction vs Full | 0.0023
2.87e-07 |
| Qwen 3 Coder | Hidden vs Interaction
Interaction vs Full | 3.54e-07
8.91e-14 |
| Claude Sonnet 4 | Hidden vs Interaction
Interaction vs Full | 6.34e-18
9.12e-06 |

Table 4: Wilcoxon signed-rank test results for Hidden vs Interaction and Interaction vs Full settings across models.

### A.4 NATURALLY UNDERSPECIFIED ISSUES

### A.5 UNDERSPECIFICITY DETECTION PROMPTS

- **Neutral**: *Ensure you have all the necessary information to proceed. If any part of the issue is unclear or lacks critical details, ask concise, targeted questions to clarify. If everything is clear, you can move ahead without asking unnecessary questions.*

- **Moderate Encouragement**: *Before attempting a solution, carefully check whether all key information is provided. If there's any ambiguity or missing details that could impact your work, don't hesitate to ask questions. Your goal is to gather the information needed for an accurate and efficient solution. Only skip asking questions when you are absolutely sure all details are complete.*

- **Strong Encouragement**: *Your success depends on having all relevant details to solve the issue effectively. Whenever you encounter unclear or missing information, proactively ask questions to fill those gaps. Even minor ambiguities can affect the outcome, so always prioritize clarifying questions. Avoid questions only when you are 100% certain no further clarification is needed.*

### A.6 QUESTION QUALITY ANALYSIS

$$\text{Cosine Distance}(P, Q) = 1 - \frac{P \cdot Q}{\|P\|\|Q\|} \tag{1}$$

where:

- $P = \{p_1, p_2, \ldots, p_N\}$ represents the embedding vector of the updated knowledge ($E_{\text{after}}$).
- $Q = \{q_1, q_2, \ldots, q_N\}$ represents the embedding vector of the initial knowledge ($E_{\text{before}}$).
- $N = 1536$ is the dimensionality of the embedding space.

| instance_id | Issue | Discussion |
|---|---|---|
| django_django-13952 | Migrate signals verbose stdout emissions are not capturable. The migrate command takes a `-verbosity` flag that is passed down to `emit_pre_migrate_signal` and `emit_post_migrate_signal` functions but these are not provided which `stdout` the output should be directed to. This makes testing `migrate -v2` through `call_command` pollute `sys.stdout` when it should be directed to the provided stdout as discovered in `https://github.com/django/django/pull/13890#pullrequestreview-579320176`. | Contains concrete technical details (function names, flags), a specific reproducibility scenario (`migrate -v2` via `call_command`), and an external reference link. |
| sympy__sympy-11794 | ASCII printing for Singularity Function. Implementation of ASCII printing for Singularity Functions is needed. | Minimal description with no code snippets or reproduction steps, showing that some natural issues are similarly vague despite lacking explicit technical context. |
| sphinx-doc__sphinx-7234 | Support for `@singledispatch` functions. It would be nice if there was some mechanism to automagically pick up the overloads to a `@functools.singledispatch` function and list them together. | Includes conversational phrasing (stream-of-thought style) and references to a specific Python mechanism, reflecting natural issue-writing patterns. |

Table 5: Examples of naturally occurring issues and their characteristic features relevant to underspecification analysis.

Table 6: Average number of questions asked per interaction.

| Model | Avg. # Questions |
|---|---|
| Claude Sonnet 4 | 4.03 |
| Qwen 3 Coder | 6.02 |
| Claude Sonnet 3.5 | 3.80 |
| Claude Haiku 3.5 | 3.49 |
| Deepseek-v2 | 4.57 |
| Llama 3.1 70B | 2.61 |

Table 7: Examples of questions and their responses across different models reflecting general patterns.

| Model | Summarized issue | Selected questions | Answers |
|---|---|---|---|
| Llama 3.1 70B | The `InheritDocstrings` metaclass fails to handle properties due to reliance on function checks that miss properties. | 1. Any known workarounds or temporary fixes?
2. Is the issue property-specific or general? | 1. I don't have that information.
2. It seems to be a general problem with how the metaclass handles properties. |
| Deepseek Chat | Same issue with property handling in `InheritDocstrings`. | 1. What is the purpose and current implementation of `InheritDocstrings`?
2. Are there existing tests that demonstrate the issue? | 1. It ensures docstrings are inherited from the parent. Uses `inspect.isfunction`, which fails for properties.
2. I don't have that information. |
| Claude Haiku 3.5 | Same issue with improper property handling by `InheritDocstrings`. | 1. Where is the metaclass located in the repo?
2. What is the specific incorrect behavior? | 1. It is in `astropy/utils/misc.py`.
2. Properties redefined without docstrings don't inherit from the base. |
| Qwen 3 Coder | Same issue with improper property handling by `InheritDocstrings`. | 1. Which classes/modules show the issue?
2. By "properties," do you mean '@property'?
3. Any tests that already capture the issue? | 1. Happens in classes using the metaclass, e.g., in 'astropy/utils/misc.py'.
2. Yes, this refers to Python's '@property'.
3. I don't have that information. |
| Claude Sonnet 3.5 | A method fails with Python 3 byte input, unlike its counterpart that handles both text and binary modes. | 1. Which method has this issue?
2. Example of a failing byte string input? | 1. The method is `Header.fromstring`.
2. No specific example, but it should contain full header data as bytes. |
| Claude Sonnet 4 | Same issue with byte input. | 1. Which class/method fails on byte-string input?
2. Example of a byte string that should work but doesn't? | 1. The failing method is 'Header.fromstring'.
2. No concrete example, just any full header as a byte string. |

## A.7 QWEN 3 CODER

---

**Example Qwen 3 thought process in Hidden setting**

**Trajectory message (verbatim):**
Let me research what the proper RFC standard is. According to RFC 2616 Section 3.3.1:

> "Recipients of timestamps SHOULD accept all three HTTP-date formats, regardless of the form in which they were sent. For the purposes of HTTP, the preferred format is IMF-fixdate. The other formats are included for compatibility with past and future implementations."

And regarding two-digit years:

> "HTTP/1.1 clients and servers that parse the date value MUST accept all three formats... All HTTP date/time stamps MUST be represented in Greenwich Mean Time (GMT), without exception. For the purposes of HTTP, GMT is exactly equal to UTC (Coordinated Universal Time)... The use of three digit years (e.g. 893) or five digit-plus years (e.g. 17763) is explicitly not allowed."

But more importantly, let me check what the actual standard practice is. Looking at common implementations and standards:

1. **POSIX/C Standard**: Often uses 69–99 $\rightarrow$ 1969–1999, 00–68 $\rightarrow$ 2000–2068
2. **RFC 2616 Recommendation**: Strongly discourages two-digit years but acknowledges backward compatibility
3. **Modern Practice**: Many systems use a 50-year sliding window or fixed windows

Let me check what the actual issue is referring to. Looking at the existing test and the problem description more carefully, I think the issue is that the current cutoff of 70 is arbitrary and doesn't follow established standards.
Let me check if there are any references or comments in the code that indicate what the intended behavior should be.

---

**Comments:** This trajectory message shows Qwen 3 Coder relying heavily on its internal knowledge of RFC standards and POSIX conventions, making specific assumptions about the intended behavior without extensive code exploration. The model directly assumes the cutoff value of 70 is problematic and implements a solution that proves correct. While this could indicate strong pretraining on software engineering standards, this pattern of solving issues through confident assertion rather than exploratory debugging is uncommon in the Hidden setting relative to models that must discover solutions through code analysis.

---

**Qwen 3 Coder and Claude Sonnet 4 Evaluation**

For both the recent models, we use OpenHands v0.60 and the updated prompts, along with an increased number of steps (up to 100) to account for the increase in capability.
For Qwen 3 Coder, we modified the interaction prompt to include a mandatory clarification step, on top of existing interaction instructions. This phase requires the model to output *only* clarifying questions and wait for responses before proceeding with the problem-solving phases. This modification was necessary because Qwen 3 Coder exhibited a rigid adherence to non-interactive SWE-Bench protocols, often bypassing interaction opportunities even when critical information was missing. The mandatory clarification phase forces the model to engage with the user before attempting implementation.

This modification ensures fair comparison in RQ1, which evaluates task success with interaction. Without it, Qwen 3 Coder defaults to non-interactive behavior, invalidating cross-model comparison. RQ2 (detection) and RQ3 (question quality) measure different capabilities and remain unaffected.

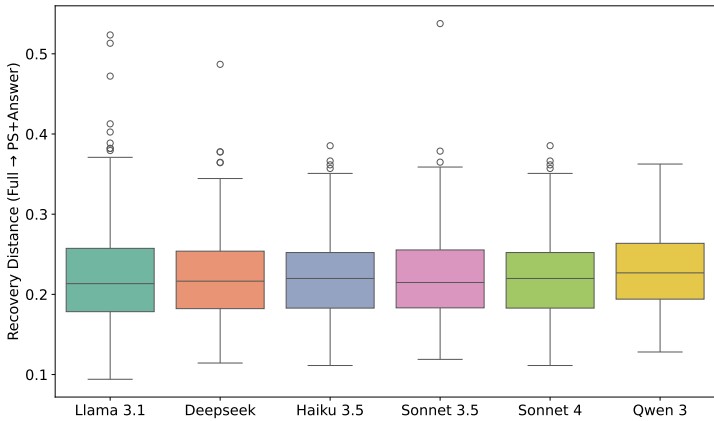

Figure 7: Recovery distance (1 - cosine similarity between full issue and post-interaction knowledge) shows minimal variation across models, failing to capture the extraction efficiency differences revealed by our original metric. This occurs because the full issue contains substantial information (formatting, links, conversational fragments) that is unnecessary for task completion. Models that ask fewer, targeted questions can obtain critical information without recovering irrelevant details, yet are penalized by this metric. In contrast, our extraction-based metric (Figure 5) better captures these differences.

## A.8    INFORMATION GAIN

