# OpenReview forum: "Ambig-SWE: Interactive Agents to Overcome Underspecificity in Software Engineering"
_ICLR.cc/2026/Conference — ICLR 2026 Poster_

### Official Review · Reviewer_osnU · 2025-10-30

**Soundness:** 3
**Presentation:** 2
**Contribution:** 3
**Rating:** 6
**Confidence:** 3

**Summary:**

The paper investigates how large language model (LLM) agents handle underspecified instructions where bug reports or feature requests omit critical details. Building on the SWE-Bench Verified dataset, the authors include three settings: (1) Full, where complete issue descriptions are given; (2) Hidden, where GPT-4o-generated summaries remove key information; and (3) Interaction, where agents can query a simulated user (GPT-4o) that has full task details under Hidden setting. The study evaluates four across three capabilities: detecting missing information, asking clarification questions, and leveraging answers to solve tasks. Results show interaction boosts success rates by up to 15.4% over non-interactive in hidden settings, though performance still lags behind fully specified inputs. Most models fail to detect underspecificity without explicit prompting, with only Claude Sonnet 3.5 achieving 84% accuracy under moderate encouragement, while Claude and DeepSeek ask more specific, information-rich questions and Llama tends to pose generic ones.

**Strengths:**

1. The paper explores a critical issue within current LLMs where they typically cannot recognize the underspecioficity in user query.
2. The paper clearly defines underspecificity as “missing information that prevents an expert from producing a correct fix,” grounding it in the SWE-Bench Verified rubric rather than using vague notions of ambiguity
3. The study divides performance into three measurable capabilities: 1) detecting underspecificity, 2) asking targeted questions, and 3) leveraging responses, which enables more diagnostic evaluation rather than a single overall score
4. Resolve-rate improvements are validated using Wilcoxon signed-rank tests to confirm significance across models.

**Weaknesses:**

1. The paper admits that naturally underspecified GitHub issues often still contain concrete technical cues (error messages, file references, conversational fragments), whereas the generated summaries mainly remove details, which may exaggerate the severity of underspecificity and may bias the task toward “missing vital context” rather than “ambiguous intent.”
2. The paper mentions using the OpenHands agent environment but gives minimal explanation of how the agent framework is structured or how its components (planning, editing, execution, and interaction) collaborate.
3. In Section 3.3, the discussion of navigational gains lacks clarity: what counts as navigational information, why it matters for task success, and whether such behavior mirrors how real developers seek information.
4. The simulated user based on GPT-4o is insufficiently validated; the paper provides no evidence that GPT-4o’s responses align with real-user behaviors or communication patterns. And there is lack of analysis about the correctness of the response to the clarification questions generated during the interaction setting.
5. In Section 5, information gain is defined as 1 – cosine similarity between the summarized task and the cumulative knowledge after interaction. This metric may overestimate improvement, as asking many irrelevant questions could still yield a high score. Would it be more accurate to calculate the score between the fully specified knowledge and the cumulative knowledge after interaction?

**Questions:**

1. How is the agent framework structured?
2. What exactly constitutes navigational information, why is it important for task success, and does this behavior reflect how real developers seek information?
3. How well does the simulated GPT-4o user align with real-user behaviors and response patterns, and how accurate are its answers to clarification questions during interaction?
4. Would it be more appropriate to compute information gain between the fully specified knowledge and the cumulative knowledge after interaction, rather than using the summarized task as the reference?

---

> ### Author Response · Authors · 2025-11-22
> **Underspecification Realism, Agent Framework Details, and Information-Gain Analysis Clarifications**
>
> Thank you for highlighting the strengths of our work, including our focus on LLMs’ difficulty with underspecificity, the clear rubric-grounded definition, the three-part diagnostic evaluation, and the statistical validation. We address the remaining concerns below.
>
> > ... Naturally underspecified GitHub issues often still contain concrete technical cues ... the generated summaries mainly remove details, which may exaggerate the severity of underspecificity and may bias the task toward "missing vital context" rather than "ambiguous intent."
>
> We agree that real issues often contain concrete technical cues that our summaries strip away, which potentially makes the underspecificity in our dataset appear more extreme than naturally occurring cases. This choice of missing vital context is by design to ensure a strong, undisputed need for interaction for task success. As we see in Table 2, LLMs still struggle to detect when information is severely missing as opposed to a complete specification. RQ2 highlights this issue in LLMs where clarification is not a primary consideration despite thorough prompting, whereas the other RQs focus on how this missing vital context is recovered. Future work on training models for interaction may focus more on natural underspecification.
>
> > The paper mentions using the OpenHands agent environment but gives minimal explanation of how the agent framework is structured or how its components (planning, editing, execution, and interaction) collaborate.
>
> We add the relevant details in the rebuttal revision in Appendix A.1. If there are other details the reviewer would like to see in the main text, please let us know.
>
> > In Section 3.3, the discussion of navigational gains lacks clarity: what counts as navigational information, why it matters for task success, and whether such behavior mirrors how real developers seek information.
>
> Thank you for this feedback. Navigational information refers to file paths requiring modification. While agents have full codebase access to discover these files independently, Table 1 shows this information substantially impacts performance for some models, revealing their code localization weaknesses. Weaker models (Llama 3.1, Deepseek-v2) frequently request file locations (30%) but fail without them (4-5% resolve rates), indicating over-reliance on explicit guidance. Our rebuttal experiments show that stronger models with better code localization (Claude Sonnet 4, Qwen 3 Coder) gain minimal benefit from navigational information (Table 1). This mirrors real developer behavior: experienced developers independently locate relevant code through search and exploration tools. Models defaulting to file location queries waste interaction turns on recoverable information rather than behavioral details that only users can provide, such as expected outcomes and error behaviors.
>
>
> > The simulated user based on GPT-4o is insufficiently validated; ... no evidence that GPT-4o's responses align with real-user behaviors ... lack of analysis about the correctness of the response to the clarification questions generated during the interaction setting.
>
> We appreciate this concern and want to clarify our design goals. Our goal is to measure models' information-gathering and integration abilities under controlled conditions, not to evaluate with realistic users. A fully realistic simulator would introduce uncontrolled variability (incorrect information, vague responses, inconsistent behavior) that confounds analysis of whether models can detect missing information, ask targeted questions, and integrate responses. This follows established practices in interactive agent evaluation [Sotopia (Zhou et al., 2024b), MediQ (Li et al., 2024), TheAgentCompany (Xu et al., 2024)]. We further limit inconsistent behavior with explicit instructions to respond "I don't have that information" for out-of-context queries. Claude Sonnet 3.5 and 4 receive nearly identical answers to similar questions (Table 7). Manual inspection confirms the proxy only provides information present in the full issue.
>
>
> > In Section 5, information gain is defined as 1 – cosine similarity between the summarized task and the cumulative knowledge after interaction ... may overestimate improvement, as asking many irrelevant questions could still yield a high score. Would it be more accurate to calculate the score between the fully specified knowledge and the cumulative knowledge after interaction?
>
> Thank you for this suggestion. We agree our current metric measures extraction volume. However, both have tradeoffs: ours cannot distinguish relevant from irrelevant questions, while the proposed metric would penalize agents that gather critical information but omit irrelevant details. We add the complementary suggested metric in Figure 7, but note that this metric fails to capture the qualitative differences observed between the models, with similar cosine distance for all models due to the above-mentioned reason.

---

> > ### Comment · Reviewer_osnU · 2025-11-24
> > **Reviewer Response**
> >
> > Thank you for addressing my concerns. My questions have been fully resolved, and I will recommend acceptance.

---

### Official Review · Reviewer_j6Aq · 2025-10-31

**Soundness:** 3
**Presentation:** 3
**Contribution:** 3
**Rating:** 6
**Confidence:** 4

**Summary:**

This paper evaluates how well LLM-based agents handle underspecified instructions in software engineering tasks. Using SWE-Bench Verified as a foundation, the authors create synthetic underspecified versions of GitHub issues and test whether models can (1) detect missing information, (2) ask effective clarifying questions, and (3) leverage interactions to solve tasks. They evaluate four models (Claude Sonnet 3.5, Claude Haiku 3.5, Llama 3.1 70B, Deepseek-v2) across three settings: fully specified issues, underspecified issues without interaction, and underspecified issues with a simulated user proxy.

**Strengths:**

1. The paper addresses a practically important problem. Real-world task descriptions are often incomplete, and understanding how agents handle this is valuable.
2. The experimental design is generally rigorous.

**Weaknesses:**

1. The most significant weakness is the lack of human validation for the synthetic underspecified issues. The authors use GPT-4o to generate summaries but provide no evidence that these summaries would actually prevent human experts from solving the tasks. Are the findings representative of real underspecification?
2. The classification of missing information into only "informational" and "navigational" details is overly simplistic. The authors mention "multiple, interdependent gaps" in real tasks but do not provide a formal taxonomy or analyze which types of underspecification are most challenging.

**Questions:**

You cite several papers on ambiguity resolution and clarification questions, but don't compare against them. Do you think those methods can be applied to the datasets for comparison?

Interactive approaches require multiple API calls, longer execution times, and user attention. Can you provide cost analysis?

---

> ### Author Response · Authors · 2025-11-22
> **Validation of Synthetic Underspecification and Missing Information Types Clarification**
>
> Thank you for the feedback! We appreciate the recognition that the paper addresses a practically important problem, and noting the rigorous experimental design. We address the concerns and questions below.
>
> > The most significant weakness is the lack of human validation for the synthetic underspecified issues. The authors use GPT-4o to generate summaries but provide no evidence that these summaries would actually prevent human experts from solving the tasks. Are the findings representative of real underspecification?
>
> We acknowledge that expert validation would be ideal. However, obtaining it at scale is prohibitively expensive. Validating 500 issues across repositories would require multiple domain experts, each needing to familiarize themselves with repository-specific conventions and attempt solutions requiring 2-4+ hours per task.
>
> Our approach provides strong proxy evidence through: (1) Distributional analysis (Section 2.1), (2) Performance gaps (3.20-30.80% Hidden vs. 8.80-68% Full) which demonstrate meaningful information loss, (3) Natural examples (Table 5) show our generations exhibit similar underspecification to real issues, and (4) LLM-annotated differences (supplementary materials) specify missing information per instance. While not perfect, these multiple validation strategies collectively support that our synthetic underspecification creates realistic and prohibitive information gaps.
>
> > The classification of missing information into only "informational" and "navigational" details is overly simplistic. The authors mention "multiple, interdependent gaps" in real tasks but do not provide a formal taxonomy or analyze which types of underspecification are most challenging.
>
> We agree that our two-way classification is coarse-grained. We intentionally use broad categories because information needs types and their importance vary significantly across issues, making the performance impact of finer taxonomies difficult to study. Our two-category split separates redundant/recoverable information (navigational), present in the codebase, from essential clarifications about the problem (informational) details. The former points to lack of model capability in code localization, as supported by Table 1. In the rebuttal revision, we see that the more recent models, Qwen 3 Coder and Claude Sonnet 4, with superior code localization abilities, do not get similar benefits from navigational information. Thus, we note that clarifications on higher-level behaviors, or informational details (e.g. error behaviors, ideal outputs) are more useful than navigational information. Developing a finer taxonomy of such information types is an important direction for future work.
>
> > You cite several papers on ambiguity resolution and clarification questions, but don't compare against them. Do you think those methods can be applied to the datasets for comparison?
>
> In the cited works, the methods developed target use cases where there is only a single point of ambiguity [1][2], which is not representative of complex real world tasks that agents undertake, or utilize smaller models not proficient on SWE-Bench tasks, preventing fair comparison.
>
> > Interactive approaches require multiple API calls, longer execution times, and user attention. Can you provide cost analysis?
>
> Because the interactive phase involved only 1–2 turns relative to the ~30 turns required for full task completion, we did not observe noticeably longer execution times. Analyzing user attention costs, such as clarification vs. feedback on an incorrect solution, is beyond the scope of this work and would be interesting future work.
>
> **References**:
>
> [1] Chen, M., Sun, R., Pfister, T., & Arık, S. Ö. (2024). Learning to clarify: Multi-turn conversations with action-based contrastive self-training (arXiv preprint arXiv:2406.00222).
>
> [2] Wang, W., Shi, J., Ling, Z., Chan, Y.-K., Wang, C., Lee, C., Yuan, Y., Huang, J.-t., Jiao, W., & Lyu, M. R. (2024). Learning to ask: When LLM agents meet unclear instruction (arXiv preprint arXiv:2409.00557).

---

### Official Review · Reviewer_x98j · 2025-11-01

**Soundness:** 3
**Presentation:** 4
**Contribution:** 3
**Rating:** 4
**Confidence:** 4

**Summary:**

The paper asks how well agents perform when faced with an underspecified or ambiguous question, where the ideal solution/user experience would require eliciting additional information from the user. The paper repurposes an existing benchmark, SWE-bench, and turn it into interactive / hidden-information benchmarks, by rephrasing and processing the inputs with LLMs. The paper asks three well-formulated research questions, which are in turned answered.

**Strengths:**

1. In my view, RQ1 and RQ2 are really interesting and well-scoped. I appreciate the experimental design, which attempts to build comparable evaluations across "Hidden", "Interaction" and "Full" settings. The primary results in Figure 3 are super interesting, and I believe potentially very influential in the field of software-engineering evaluations. However, I would really like to see a more comprehensive set of models here.
2. The construction of the dataset is well-described and I appreciate the careful analysis in section 2.1 that elicits qualitative differences between human-written underspecified problem statements and the synthetically rewritten issues.

**Weaknesses:**

1. Outdated models: Both proprietary (Claude-3.5) and open-source models (DeepSeek-v2 and Llama-70b) are unfortunately rather behind the state of the art, given the rate of progress in recent months. Just for Claude models alone, we've seen Sonnet-3.7, Sonnet-4, and Sonnet-4.5 in the meantime. For this paper to be relevant for a conference presentation, I fear that the paper would really require updated results from more up-to-date frontier models. This will also make the claims around open-weight vs. proprietary models more relevant. In fact, I am intruiged if the type of "interaction" gap in the paper still remains with the current generation of models.

2. I find the results and methodology for RQ3 "Can the LLM generate meaningful and targeted clarification questions?" slightly underwhelming, since the main message seems to be that Llama 3.1 70b is not very good at question answering. The evaluation metrics do not really seem to be able to reveal fine-grained differences in question-asking abilities. The qualitative analysis is nice, but in my opinion lacks comprehensive documentation of full problem statements and agent traces featured in the appendix.

3. The paper relies quite heavily on SWE-bench for the original problem statements, which is quite heavily targeted by model creators. I think an analysis of how results would transfer to a more niche software engineering settings would be very interesting.

**Questions:**

> We did not evaluate on naturally underspecified SWE-Bench examples because they lack the paired ground truth (complete specifications) necessary for causal measurement of interaction impact.

Would it be possible to infer the complete specification from the known `gold_patch` and `test_patch`? How would this approach compare to the chosen approach in the paper?

> However, there are naturally occurring underspecified issues that are similarly vague as well (django_django-13952, django_django-15744, pytest-dev_pytest-7283, sphinx-doc_sphinx-9467, sympy_sympy-12977 are some specific examples)

I would encourage the authors to feature and discuss these examples in the appendix, and simply refer to the appendix here. This is a rather distracting amount of information for page 3.

---

> ### Author Response · Authors · 2025-11-22
> **Updated Models, RQ3 Clarifications**
>
> Thank you for the detailed review. We appreciate the positive feedback regarding our experimental and dataset design, as well as the assessment of the work’s potential influence. We address the raised concerns below.
>
>
> > Both proprietary (Claude-3.5) and open-source models (DeepSeek-v2 and Llama-70b) are ... behind the state of the art... updated results ... make the claims around open-weight vs. proprietary models more relevant. In fact, I am intruiged if the type of "interaction" gap in the paper still remains with the current generation of models.
>
> Thank you for raising this concern. We have conducted additional experiments across all 3 RQs and settings with two recent models, both proprietary and open-weight (**Claude Sonnet 4** and **Qwen 3 480B coder**), and find three key results:
>
> 1. The open-weight model Qwen 3 achieves **46%** resolve rate in the Interaction setting, even exceeding Claude Sonnet 4's **41.8%**, demonstrating that the interaction gap between proprietary and open-weight models is closing.
>
> 2. Claude Sonnet 4 shows diminishing returns from interaction compared to Claude Sonnet 3.5 (61% vs 80% full performance recovery), despite achieving significantly higher absolute performance in the Full setting (68% vs 49.4%).
>
> 3. Qwen 3 requires stronger interaction prompts even with mandatory interaction than other models to engage with users (Appendix A.7), otherwise defaulting to non-interactive behavior across all prompts (Table 2).
>
> These evolving patterns across model generations underscore the value of our benchmark for tracking how models’ interaction strategies change with scale and capability.
>
> >  I find the results and methodology for RQ3 "Can the LLM generate meaningful and targeted clarification questions?" slightly underwhelming... The evaluation metrics do not really seem to be able to reveal fine-grained differences in question-asking abilities...
>
> Thank you for the feedback. We have revised RQ3 to link how the similar quantitative performance of some models is explained by the differing qualitative aspects of the questions posed. With the more recent models, we also observe improvements in the quantitative metrics, showing that the metrics are capable of capturing fine-grained differences. We identify 3 key behaviors that impact the information gain in Figure 5:
>
> 1. **Quantity-quality tradeoff**: Qwen 3 asks the most questions (6 per interaction) for slightly higher information gain (0.179 cosine distance), while Claude Sonnet 4 achieves comparable gain (0.171) with fewer questions (4 on average), showing efficiency.
> Deepseek and Claude Sonnet 3.5 show a similar tradeoff at similar cosine distances (~0.140).
>
> 2. **Exploration-first vs. question-first**: Claude models often explore the codebase before asking questions, enabling targeted inquiries, whereas others clarify immediately, often focusing on recoverable details like file locations.
>
> 3. **Answerability vs. specificity**: Deepseek’s highly specific questions (low level implementation details) often receive denial from the user, wasting turns; Claude models generally balance specificity with assumptions about user knowledge.
>
> We have expanded appendix examples (Table 7) to illustrate these patterns.
>
> > The paper relies quite heavily on SWE-bench ... heavily targeted by model creators. I think an analysis of how results would transfer to a more niche software engineering setting would be very interesting.
>
> We agree this analysis is important, and would be interesting for future work. We can add discussion on this in the next revision.
>
> > Would it be possible to infer the complete specification from the known gold_patch and test_patch? How would this approach compare to the chosen approach in the paper?
>
> The test_patch and gold_patch capture what makes a solution correct, but cannot reveal which parts of the issue description were critical for understanding the problem. While there is a clear mapping from tests to code (confirming fixes work), there is no direct mapping from tests back to the conceptual insights that motivated the solution. Thus it is difficult to infer the complete specification from these. Our approach, in comparison, focuses on pairing the underspecified issues with the fully specified versions such that the user has access to the full information of the desired solution.
>
> > However, there are naturally occurring underspecified issues that are similarly vague as well (django_django-13952, django_django-15744, pytest-dev_pytest-7283, sphinx-doc_sphinx-9467, sympy_sympy-12977 are some specific examples). I would encourage the authors to feature and discuss these examples in the appendix, and simply refer to the appendix here. This is a rather distracting amount of information for page 3.
>
> We have moved this discussion to Appendix A.4 in the rebuttal revision.
>
> **Thank you again for your constructive feedback. We believe these revisions have substantially strengthened the paper.**

---

> > ### Comment · Reviewer_x98j · 2025-11-22
> > **Reviewer response**
> >
> > Thank you for addressing my concerns. I will raise my score and recommend acceptance.

---

### Meta-Review · Area_Chair_yRgj · 2026-01-06

**Summary:**

Overall, this paper makes a solid and timely contribution by isolating underspecificity in SWE agent benchmarks and proposing a controlled evaluation protocol on SWE-Bench Verified via Full / Hidden / Interaction settings, where interaction is enabled through a user proxy holding the full issue context. The results support the central claim that interaction can materially improve performance, while also highlighting that reliably detecting when key information is missing remains a major bottleneck.

**Reviewer Concerns:**

**Reviewer x98j**'s main concerns (model staleness and an underwhelming RQ3 story) were largely addressed in the rebuttal via added experiments with newer models (e.g., Claude Sonnet 4 and Qwen 3 Coder), a clearer interpretation of RQ3, and some reorganization of distracting material; the reviewer explicitly indicated their concerns were resolved and recommended acceptance.

**Reviewer j6Aq** focused on external validity—especially the lack of human validation for synthetic underspecification—and on the coarseness of the information-type taxonomy and limited cost/baseline comparisons; the rebuttal provides reasonable proxy evidence and clarifications, but the human-validation and richer-taxonomy limitations remain partially outstanding.

**Reviewer osnU**'s questions about realism, framework details, the meaning of “navigational” information, the simulated user, and the information-gain metric were addressed with clarifications and additions, and the reviewer explicitly indicated their concerns were resolved and recommended acceptance.

**Reviewer Scores:**

In my opinion, after normal rebuttal phase, Reviewer j6Aq would maintain its score, while Reviewer x98j and Reviewer osnU would increase the score, making the paper to be accept.

---

### Decision · Program_Chairs · 2026-01-26

Accept (Poster)